# Bedside clinical prediction tool for mortality in critically ill children

**Kanokkarn Sunkonkit**[1,2,3], **Chatree Chai-adisaksopha**[2,3,4], **Rungrote Natesirinilkul**[5], **Phichayut Phinyo**[2,3], **Konlawij Trongtrakul**[2,3,6]*

1 Department of Pediatrics, Division of Pulmonary and Sleep Medicine, Faculty of Medicine, Chiang Mai University, Chiang Mai, Thailand, 2 Center for Clinical Epidemiology and Clinical Statistics, Faculty of Medicine, Chiang Mai University, Chiang Mai, Thailand, 3 Department of Biomedical Informatics and Clinical Epidemiology (BioCE), Faculty of Medicine, Chiang Mai University, Chiang Mai, Thailand, 4 Department of Internal Medicine, Division of Hematology, Faculty of Medicine, Chiang Mai, Thailand, 5 Department of Pediatrics, Division of Hematology and Oncology, Faculty of Medicine, Chiang Mai University, Chiang Mai, Thailand, 6 Department of Internal Medicine, Division of Pulmonary, Critical Care, and Allergy, Faculty of Medicine, Chiang Mai University, Chiang Mai, Thailand

* konlawij.tr@cmu.ac.th

## Abstract

### Introduction

Mortality rates among critically ill pediatric patients remain a persistent challenge. It is imperative to identify patients at higher risk to effectively allocate appropriate resources. Our study aimed to develop a prediction score based on clinical parameters and hemogram to predict pediatric intensive care unit (PICU) mortality.

### Methods

We conducted a retrospective study to develop a clinical prediction score using data from children aged 1 month to 18 years admitted for at least 24 hours to the PICU at Chiang Mai University between January 2018 and December 2022. PICU mortality was defined as death within 28 days of admission. The score was developed using multivariable logistic regression and assessed for calibration and discrimination.

### Results

There were 29 deaths in 330 children (8.8%). Our model for predicting 28-day ICU mortality uses four key predictors: male gender, use of vasoactive drugs, red blood cell distribution width (RDW) ≥15.9%, and platelet distribution width (PDW), categorized as follows: <10% (0 points), 10–14.9% (2 points), and ≥15% (4 points). Scores range from 0 to 8, with a cutoff value of 5 to differentiate low-risk (<5) from high-risk (≥5) groups. The tool demonstrates excellent performance with an AuROC curve of 0.86 (95% CI: 0.80–0.91, p<0.001) showing excellent discrimination and calibration, 82.8% sensitivity, and 73.1% specificity, respectively.

**Data availability statement:** All relevant data are within the manuscript and its Supporting Information files.

**Funding:** This work was supported by the Faculty of Medicine Research Fund, Chiang Mai University, Chiang Mai, Thailand (Grant No. 119/2566). The funders had no role in study design, data collection and analysis, decision to publish, or preparation of the manuscript.

**Competing interests:** The authors have declared that no competing interests exist.

## Conclusions

The score, developed from clinical data and hemogram, demonstrated potential in predicting ICU mortality among critically ill children. However, further studies are necessary to externally validate the score before it can be confidentially implemented in clinical practices.

## Introduction

Despite advances in medical technology, pediatric intensive care unit (PICU) mortality rates vary widely between developed and developing countries. The PICU mortality rates in developed countries typically range between 4.0% to 7.3%. However, these rates are significantly higher in developing countries, reaching as high as 13.0% to 25.0% [1,2]. This disparity reflects differences in healthcare infrastructure, resource availability, and socioeconomic status that impact patient outcomes.

To optimize the care of critically ill pediatric patients, it is crucial to prioritize "at risk" patients for appropriate resources, enabling timely intervention and optimal management. A reliable scoring system can be a vital tool for predicting PICU outcomes and guiding physicians in making decisions about resource allocation, particularly in settings with limited resources.

The current landscape of mortality prediction scoring systems in PICUs presents significant challenges. Widely used systems like Pediatric Risk of Mortality (PRISM) III/IV, Pediatric Index of Mortality (PIM)-2 and -3, and Pediatric Logistic Organ Dysfunction-2 (PELOD-2) [3,4] have limitations in their application [3]. This prediction scoring system relies on arterial blood gas parameters such as pH, partial pressure of arterial oxygen ($PaO_2$), partial pressure of arterial carbon dioxide ($PaCO_2$), and base excess for score calculation. However, obtaining arterial blood gas measurements can be challenging, particularly in young children and infants. Additionally, serum biomarkers such as serum ferritin, lactate, C-reactive protein (CRP), and microalbuminuria are commonly employed in assessing the necessity for admission and prognostication [5–8]. However, their utility is often restricted by cost and availability, making them feasible primarily in tertiary-care or university hospitals or in developed countries.

Complete blood count (CBC) is inexpensive and widely available laboratory test and measure characteristics of red blood cell (RBC), white blood cell (WBC) and platelets. Several studies showed that RBC and platelet parameters including red blood cell distribution width (RDW), mean platelet volume (MPV) and platelet distribution width (PDW) were associated with inflammatory diseases and could predict ICU mortality [9–15].

Therefore, the aim of our study was to develop a prediction score to determine ICU mortality in critically ill pediatric patients, using clinical and hemogram parameters. This approach is designed to enhance applicability, especially in primary-care hospitals, by facilitating prompt bedside assessment and decision-making.

## Materials and methods

### Study design and setting

This study was conducted retrospectively from critically pediatric patients who were admitted to two PICUs of the Faculty of Medicine, Chiang Mai University, Thailand, between January 2018 and December 2022. The study protocol was approved by the Research Ethics committee of the Faculty of Medicine, Chiang Mai University (approval no. 166/2566 on 10 May 2023) and was performed in accordance with the Declaration of Helsinki as a statement of ethical principles for medical research involving human subjects. Informed consent was waived due to minimal risk, and the data were analyzed anonymously. The data were accessed for research purpose during 1 June 2023 to 31 December 2023.

### Eligibility criteria

**Inclusion criteria.** All children aged 1 month to 18 years who admitted to our PICUs were included. These patients were required to have a CBC upon PICU admission, and their ICU stay had to be at least 24 hours. Subsequent readmissions were considered new episodes of admissions.

**Exclusion criteria.** We excluded patients with the following conditions: children who had received a RBC transfusion within 21 days or a platelet transfusion within 7 days prior to the PICU admission; children diagnosed with beta-thalassemia major, beta-thalassemia/hemoglobin E (Hb E), and hemoglobin H (Hb H) disease, as these conditions can affect RDW; children diagnosed with congenital hemolytic anemia, such as red blood cell membrane defect, that impact RDW; and children diagnosed with congenital macrothrombocytopenia or myeloproliferative disorders, which can affect platelet count, MPV and PDW.

### Data collection

We collected information on study participants, including patient characteristics such as age, sex, body weight, height, body mass index (BMI), reasons for ICU admission, pre-existing comorbidities, thalassemia trait, and iron deficiency anemia. ICU parameters were also recorded, including the use of mechanical ventilation (MV), duration of MV support, use of vasoactive drugs, use of continuous renal replacement therapy (CRRT), presence of multi-organ dysfunction (MOD), PIM-2 score, and ICU length of stay (ICU-LOS). Additionally, CBC results from within the first 24 hours of ICU admission were included in the analysis. The CBCs were assessed using the Sysmex XN-9000® analyzer (Sysmex Co., Kobe, Japan).

### Candidate predictors

We identified 12 potential predictors for developing the ICU mortality prediction score based on prior knowledge, clinical expertise, and a comprehensive literature review. These predictors included age [16–18]; sex [18,19]; BMI [20]; comorbidities [21]; use of MV [22]; use of vasoactive drug [18,22,23]; presence of MOD [16,22]; RDW [9–12,24–28]; WBC [18,25]; platelet count [15,18,25,29]; MPV [15,30]; and PDW [15]. The prediction score for our cohort was calculated using data collected within the first 24 hours of ICU admission.

### Study endpoint

The study endpoint was 28-day ICU mortality, defined as death occurring within 28 days of PICU admission.

### Study size estimation

The study's sample size was determined by examining factors predicting mortality in the PICU, based on a pilot study of 256 patients conducted at Chiang Mai University Hospital from June 2012 to June 2013. This pilot study found a mortality

rate of 14.5%, with 219 survivors and 37 non-survivors. Sample size calculations were performed using two-sample proportion comparisons related to PICU mortality, specifically looking at thrombocytopenia (34.2% in survivors vs. 86.5% in non-survivors) and MV use (72.1% in survivors vs. 94.6% in non-survivors). With a significance level (alpha) of 0.05 (two-sided) and a desired power of 0.80, the estimated sample sizes needed were 64 based on thrombocytopenia and 218 based on MV support. To address potential data loss, we adjusted for 25% missing values, ensuring a minimum sample size of 290. We calculated the necessary sample size to develop a prediction model aimed at accurately estimating the population's average outcome risk at a specific time point of interest. Based on an expected outcome incidence of 8%, the minimum required sample size for model development was determined to be 114, representing 10 events. To enhance precision, we plan to include all eligible patients within the study period [31].

## Statistical analysis

Descriptive statistics were utilized to summarize the study population, including frequencies, percentages, and median and interquartile range (IQR), as appropriate. Univariable and multivariable logistic regression analyses, along with the calculation of the area under the receiver operating characteristic (AuROC) curve, were conducted using STATA Statistical software version 17 (StataCorp, College Station, Texas). Statistical significance was set at a p-value of less than 0.05.

## Prediction score development

We tested for multicollinearity and included pre-selected predictors in the analytic model. Both univariable and multivariable logistic regression analyses were performed to explore the relationship between the independent predictors and 28-day ICU mortality. The initial model included all predictors, but backward elimination was applied to exclude those predictors with a p-value greater than 0.05. Given the 29 non-survivors in our cohort, a maximum of 4 variables was used to develop the prediction score. After model reduction, the point for each predictor was calculated by dividing each regression coefficient by the smallest regression coefficient, and rounding to the nearest integer. The final score was then obtained by summing these rounded points.

We divided the scores into two clinical risk categories for practical application: low-risk and high-risk. A score below 5 points indicates a lower risk of mortality, whereas a score of 5 or higher signifies a substantially increased risk. We also assessed and compared the diagnostic performance of the score and the PIM-2 score, examining sensitivity, specificity, positive predictive value (PPV), negative predictive value (NPV), and likelihood ratios through statistical analysis and graphical representation.

The predictive performance of the score was evaluated through calibration and discrimination, with calibration assessed using the Hosmer–Lemeshow goodness-of-fit test and a calibration plot, while discriminative ability was measured with the area under the AuROC. For internal validation, bootstrap resampling with 300 replicates was employed to estimate the model's optimism.

## Results

### Study population

Four hundred and fifty-three pediatric patients were admitted to the ICU. Of these, 123 were excluded due to recent transfusions: 87 had received RBC transfusions within the preceding 21 days, 1 had received a platelet transfusion within the last 7 days, and 35 had received both. Consequently, 330 children (49.7% male) were included in the study. The median age of the participants was 4.33 (IQR: 2.91–8.67) years, and their median BMI was 14.60 (IQR: 12.99–16.56) kg/m². The most common reasons for ICU admission were post-operative care (40.3%), pneumonia with respiratory failure (32.4%), septic shock (17.6%), and heart failure (17.6%), respectively. Intubation was required for 248 children (75.2%), and 152 children (46.1%) required vasoactive drugs administration within the first 24 hours of ICU admission. The median length

of stay in the PICU was significantly longer in the non-survival group compared to the survival group (8 days [IQR: 3–26 days] vs. 5 days [IQR: 3–11 days], p<0.001). The overall PICU mortality rate was found to be 8.8%. Detailed demographic characteristics of the study population are presented in Table 1.

Continuous data are presented as mean ± SD. Otherwise (*) denotes a report as median and interquartile range (IQR). Other reasons for ICUs admission: severe asthma exacerbation, acute myocarditis, cardiogenic shock, arrhythmia, hypoxic spell, seizure, diabetes ketoacidosis, hypovolemic shock and CRRT or plasmapheresis. Other comorbidities: gastrointestinal disease, endocrine disease and nephrological disease.

The factors associated with mortality in critically ill children are demonstrated in Table 2 and 3. According to multivariable logistic regression analysis, the only factors significantly associated with mortality in critically ill children were male gender (mOR = 2.70; 95% CI 1.07–6.79, $p= 0.034$), use of vasoactive drugs (mOR = 3.69; 95% CI 1.40–9.72, $p= 0.008$), RDW≥15.9% (mOR = 4.08; 95% CI 1.43–11.61, $p= 0.008$), PDW 10–14.9% (mOR = 5.79; 95% CI 1.29–26.01, $p= 0.022$), and PDW ≥15% (mOR =34.72; 95% CI 6.28–191.98, $p<0.001$), respectively. Consequently, these parameters have led to the development of the predictive score for mortality in critically ill children, termed the "**E**arly **S**creening **C**ritically **I**ll **C**hildren (ESCIC) score". The ESCIC score ranges from 0 to 8.

The discrimination capacity of the ESCIC score, as determined by AuROC curve, was 0.86 (95% CI: 0.80–0.91), while that of the PIM-2 was 0.64 (95% CI: 0.53–0.74). The ESCIC score significantly improved AuROC performance (p < 0.001), as illustrated in Fig 1A and 1B. Internal validation of the ESCIC score, conducted through a bootstrapping procedure with 300 replicates, revealed an apparent AuROC of 0.86 (range: 0.80–0.91) and a test AuROC of 0.85 (range: 0.78–0.91). The optimism of the AuROC was calculated to be 0.01.

For practical application, the derived models are presented as the ESCIC, based on individual inputs. This application estimates the predicted probability of 28-day mortality in critically ill children. We categorized the scores into 2 clinical risk categories: low-risk and high-risk. An ESCIC score below 5 points indicates a low probability of mortality, while a score of 5 or higher signifies a substantially increased probability of mortality. The sensitivity and specificity levels of the ESCIC score were 82.8% (95% CI: 64.2–94.2) and 73.1% (95% CI: 67.7–78.0), respectively in Table 4.

## Discussion

In reality, mortality rates among critically ill children are influenced by various factors, such as case-mix, infrastructure, environmental hygiene, personnel expertise, facilities, and economic policies within the country. Given the advancements in pediatric care within the ICU, maintaining strict quality control is crucial to identify high-risk groups for mortality and optimize treatment adequacy, as well as to plan and efficiently utilize resources. Variations in mortality rates across PICUs often arise from differences in illness severity.

In this context, our study introduces an innovative prediction model termed the "ESCIC score," designed for mortality screening in critically ill children. Our model integrates four readily available predictors—male gender, use of vasoactive drugs, RDW, and PDW—into a comprehensive risk assessment tool. The ESCIC score demonstrated robust discriminatory performance and excellent calibration. By incorporating these key variables, the ESCIC score provides a valuable tool for risk stratification and informed clinical decision-making within pediatric intensive care settings.

In our study, we observed a higher mortality rate among male patients in the PICUs. This finding aligns with existing literature that suggests gender differences in critical care outcomes [32–34]. However, the reasons behind increased mortality in male PICU patients are multifaceted. Biological factors, such as differences in immune responses between genders, may play a significant role. Additionally, behavioral and social factors could contribute to the observed disparities. It's important to note that findings on gender differences in PICU mortality are not entirely consistent across studies. Given these complexities, further research is necessary to elucidate the underlying mechanisms contributing to gender disparities in PICU mortality. Understanding these factors is crucial for developing targeted interventions aimed at improving outcomes for all critically ill children.

**Table 1. Clinical characteristics and laboratory parameters of children who were admitted to the PICUs (n = 330).**

| Characteristics | Non-survivors (n= 29) | Survivors (n= 301) | Univariable analysis OR (95% CI) | P-value | AUROC |
|---|---|---|---|---|---|
| **Clinical characteristics** | | | | | |
| **Age (years)*** | 3.8 (2.8, 7.2) | 4.3 (3.0, 8.7) | 0.97 (0.88-1.06) | 0.53 | 0.54 (0.43-0.64) |
| **Age <24 months, n (%)** | 1 (3.4) | 36 (11.9) | 3.80 (0.50-28.81) | 0.19 | 0.54 (0.50-0.58) |
| **Male, n (%)** | 20 (68.9) | 144 (47.8) | 2.42 (1.06-5.49) | 0.034 | 0.60 (0.51-0.69) |
| **BMI (kg/m²)*** | 14.3 (12.9, 18.7) | 14.6 (13.0, 16.5) | 1.05 (0.95-1.16) | 0.32 | 0.52 (0.40-0.64) |
| **Reason for PICUs admission** | | | | | |
| **Pneumonia with respiratory failure, n (%)** | 14 (48.3) | 93 (30.9) | 2.08 (0.96-4.50) | 0.06 | 0.59 (0.49-0.68) |
| **ARDS, n (%)** | 3 (10.3) | 5 (1.7) | 6.83 (1.54-30.20) | 0.011 | 0.54 (0.49-0.60) |
| **Septic shock, n (%)** | 12 (41.4) | 46 (15.3) | 3.91 (1.75-8.73) | 0.001 | 0.63 (0.54-0.72) |
| **Heart failure, n (%)** | 8 (27.6) | 50 (16.6) | 1.91 (0.80-4.56) | 0.14 | 0.55 (0.47-0.64) |
| **Postcardiac arrest, n (%)** | 2 (6.9) | 16 (5.3) | 1.32 (0.29-6.04) | 0.72 | 0.51 (0.46-0.56) |
| **Upper GI hemorrhage, n (%)** | 3 (10.3) | 7 (2.3) | 4.84 (1.18-19.86) | 0.028 | 0.54 (0.48-0.60) |
| **Post-operative care, n (%)** | 5 (17.2) | 128 (42.5) | 0.28 (0.10-0.76) | 0.012 | 0.62 (0.55-0.70) |
| **Others, n (%)** | 6 (20.7) | 69 (22.9) | 0.88 (0.34-2.24) | 0.78 | 0.51 (0.43-0.59) |
| **Comorbidities** | | | | | |
| **Chronic lung disease, n (%)** | 1 (3.4) | 15 (4.9) | 0.68 (0.09-5.34) | 0.71 | 0.51 (0.47-0.54) |
| **Congenital heart disease, n (%)** | 19 (65.5) | 140 (46.5) | 2.18 (0.98-4.85) | 0.06 | 0.59 (0.50-0.69) |
| **Solid tumor/ leukemia, n (%)** | 4 (13.8) | 13 (4.3) | 3.54 (1.07-11.68) | 0.038 | 0.55 (0.48-0.61) |
| **Others, n (%)** | 4 (13.8) | 22 (7.3) | 2.03 (0.65-6.35) | 0.22 | 0.53 (0.46-0.59) |
| **Laboratory parameters** | | | | | |
| **Hemoglobin (g/dl)** | 11.7±3.4 | 11.5±2.7 | 1.03 (0.89-1.17) | 0.69 | 0.50 (0.38-0.63) |
| **Hematocrit (%)** | 35.9±12.0 | 34.6±8.4 | 1.02 (0.97-1.06) | 0.42 | 0.51 (0.39-0.64) |
| **MCV (fl)** | 80.7±6.9 | 79.0±8.2 | 1.03 (0.98-1.08) | 0.29 | 0.54 (0.44-0.65) |
| **RBC count (cell/mm³)** | 4.5±1.4 | 4.4±1.1 | 1.08 (0.77-1.50) | 0.66 | 0.51 (0.38-0.64) |
| **RDW (%)** | 19.2±3.9 | 16.6±3.4 | 1.17 (1.08-1.28) | <0.001 | 0.72 (0.62-0.81) |
| **RDW≥15.9%, n (%)** | 24 (82.8) | 153 (50.8) | 4.64 (1.72-12.49) | 0.002 | 0.55 (0.52-0.58) |
| **White blood cell count (cell/mm³)*** | 11720 (7890, 19710) | 12180 (8680, 16630) | 1.00 (0.99-1.00) | 0.26 | 0.51 (0.38-0.64) |
| **Platelet count (cell/mm³)*** | 139000 (86000, 265000) | 232000 (142000, 340000) | 0.99 (0.99-1.00) | 0.010 | 0.67 (0.55-0.78) |
| **MPV (fl)** | 11.2±1.3 | 9.8±1.0 | 2.71 (1.89-3.88) | <0.001 | 0.82 (0.75-0.88) |
| **MPV≥10.25 fl, n (%)** | 20 (68.9) | 82 (27.2) | 5.93 (2.59-13.56) | <0.001 | 0.57 (0.54-0.62) |
| **PDW (%)** | 13.8±2.2 | 10.7±2.1 | 1.69 (1.42-2.01) | <0.001 | 0.85 (0.78-0.91) |
| **PDW < 10%, n (%)** | 2 (6.9) | 133 (44.2) | Reference | | 0.75 (0.68-0.83) |
| **PDW 10–14.9%, n (%)** | 18 (62.1) | 155 (51.5) | 7.72 (1.76-33.89) | 0.007 | |
| **PDW ≥15.0%, n (%)** | 9 (31.0) | 13 (4.3) | 46.03 (8.97-236.06) | <0.001 | |
| **Creatinine (mg/dL)*** | 0.4 (0.2, 0.9) | 0.3 (0.2, 0.5) | 1.04 (0.85-1.25) | 0.69 | 0.52 (0.39-0.66) |
| **Albumin (g/dL)*** | 3.5 (3.1, 4.0) | 3.4 (3.0, 3.8) | 1.35 (0.68-2.71) | 0.39 | 0.54 (0.42-0.67) |
| **AST (U/L)*** | 111 (50, 197) | 43.5 (27, 87) | 1.00 (1.00-1.00) | 0.002 | 0.70 (0.59-0.81) |
| **ALT (U/L)*** | 42 (29, 167) | 24 (13, 48) | 1.00 (1.00-1.00) | 0.015 | 0.71 (0.61-0.81) |
| **Total bilirubin (mg/dL)*** | 0.8 (0.3, 2.5) | 0.5 (0.3, 1.0) | 1.10 (0.98-1.24) | 0.11 | 0.62 (0.49-0.74) |
| **Direct bilirubin (mg/dL)*** | 0.6 (0.2, 1.0) | 0.3 (0.2, 0.5) | 1.19 (1.01-1.43) | 0.043 | 0.65 (0.53-0.77) |

*(Continued)*

**Table 1.** (Continued)

| Characteristics | Non-survivors (n= 29) | Survivors (n= 301) | Univariable analysis | | AUROC |
|---|---|---|---|---|---|
| | | | OR (95% CI) | *P-value* | |
| **Severity of illness** | | | | | |
| PIM-2 score* | 2.4 (1.3, 7.6) | 1.8 (0.8, 4.4) | 1.04 (1.01-1.07) | 0.010 | 0.64 (0.53-0.74) |
| ESCIC score* | 5.6 (5.0, 6.0) | 3.1 (2.0, 5.0) | 2.41 (1.77-3.27) | <0.001 | 0.86 (0.80-0.91) |
| **Management** | | | | | |
| Use of mechanical ventilation, n (%) | 25 (86.2) | 223 (74.1) | 2.19 (0.74-6.48) | 0.15 | 0.56 (0.49-0.63) |
| Duration of mechanical ventilation support (days)* | 10 (1, 13) | 3 (0, 8) | 1.05 (1.02-1.08) | 0.001 | 0.63 (0.52-0.76) |
| Use of vasoactive drugs, n (%) | 22 (75.9) | 130 (43.2) | 4.13 (1.71-9.97) | 0.002 | 0.66 (0.58-0.75) |
| Vasoactive inotropic score within the first 24 hours post-admission* | 120 (10, 150) | 0 (0, 10) | 1.06 (1.03-1.07) | <0.001 | 0.81 (0.69-0.92) |
| Multi-organ failure, n (%) | 21 (72.4) | 145 (48.2) | 2.82 (1.21-6.57) | 0.016 | 0.62 (0.53-0.71) |
| Number of organ dysfunction* | 4 (0, 4) | 0 (0, 1) | 4.04 (2.69-6.04) | <0.001 | 0.77 (0.65-0.89) |

**Table 2.** Multivariable full model of factors associated with 28-day ICU mortality in critically ill children (n = 330).

| Variables within 24-hour of admission | | Full model (n= 330) | |
|---|---|---|---|
| | | OR (95% CI) | *P-value* |
| Age < 24 months | | 0.26 (0.03-2.59) | 0.254 |
| Male | | 2.98 (1.13-7.88) | 0.028 |
| BMI (kg/m²) | | 1.02 (0.88-1.18) | 0.748 |
| Comorbidities | None | Reference | |
| | 1 | 0.41 (0.12-1.39) | 0.153 |
| | ≥ 2 | 1 (omitted) | |
| RDW≥15.9% | | 4.86 (1.56-15.08) | 0.006 |
| WBC | | 1.00 (0.99-1.00) | 0.153 |
| Platelet count | | 0.99 (0.99-1.00) | 0.165 |
| MPV≥10.25 fl | | 2.49 (0.83-7.49) | 0.103 |
| PDW | < 10% | Reference | |
| | 10-14.9% | 4.95 (0.91-26.77) | 0.063 |
| | ≥15.0% | 16.65 (1.99-138.66) | 0.009 |
| Use of mechanical ventilation | | 2.22 (0.55-9.00) | 0.265 |
| Use of vasoactive drugs | | 4.25 (1.31-13.72) | 0.015 |
| Multi-organ failure | | 0.77 (0.25-2.44) | 0.664 |

In our study, we observed that three-quarters of non-surviving children were prescribed vasoactive drugs. Consistently, numerous studies have demonstrated that the use of vasoactive drugs strongly correlates with in-ICU death, indicating a higher ICU mortality rate among patients who received such medications [35,36]. The administration of vasoactive drugs in pediatric patients is associated with increased mortality [22,23]. This correlation may stem from the fact that children prescribed these medications often exhibit unstable hemodynamics, highlighting the severity of their condition. Vasoactive drugs play a crucial role in managing shock, with catecholamines being the most commonly utilized agents in the ICU [37,38].

**Table 3. Final model of factors that are associated with ICU mortality in critically ill children (n = 330).**

| Variables within 24-hour of admission | Reduced model (n=330) | | β | P-value | Assigned score |
|---|---|---|---|---|---|
| | OR (95% CI) | P-value | | | |
| Male | 2.70 (1.07-6.79) | 0.034 | 0.99 | 0.034 | 1 |
| Use of vasoactive drugs | 3.69 (1.40-9.72) | 0.008 | 1.30 | 0.008 | 1 |
| RDW≥15.9% | 4.08 (1.43-11.61) | 0.008 | 1.40 | 0.008 | 1 |
| PDW | | | | | |
| < 10% | Reference | | Reference | | 0 |
| 10-14.9% | 5.79 (1.29-26.01) | 0.002 | 1.76 | 0.002 | 2 |
| ≥15.0% | 34.72 (6.28-191.98) | <0.001 | 3.54 | <0.001 | 4 |

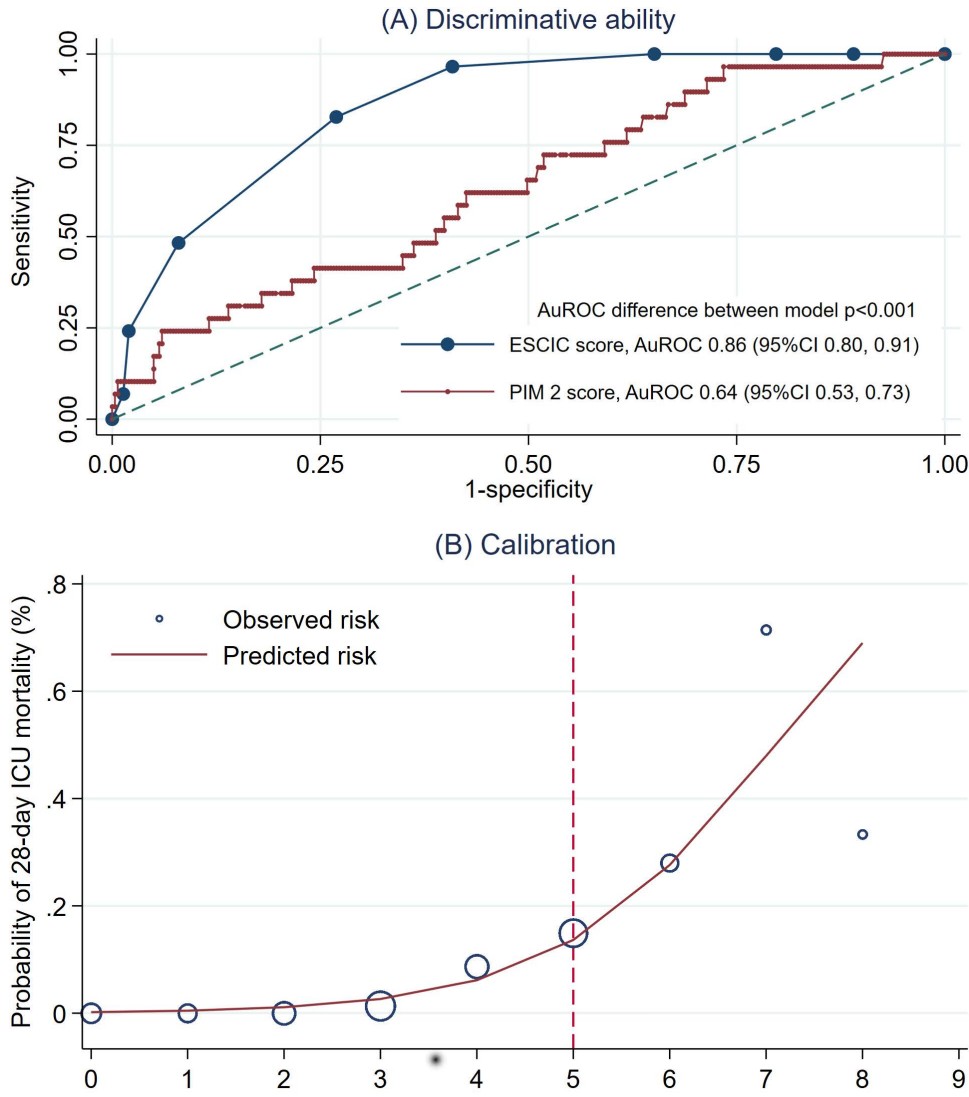

**Fig 1. Discrimination and calibration performance of Early Screening Critically Ill Children (ESCIC) score.** (A) Area under the receiver operating characteristics (AuROC) curve of Early Screening Critically Ill Children (ESCIC) score compared to PIM-2 score. (B) Model calibration plot of agreement between model predicted odds and observed proportion of children with mortality.

**Table 4. Prediction score for anticipating ICU mortality in critically ill children (n = 330).**

| ESCIS score cut point score | ESCIC score | Probability of mortality | Non-survivors (n=29) | Survivors (n=301) | Sensitivity | Specificity | PPV | NPV | LR+ (95%CI) | LR- (95%CI) |
|---|---|---|---|---|---|---|---|---|---|---|
| 4 | < 4 | Low | 1 (3.45) | 178 (59.1) | 96.6 (82.2-99.9) | 59.1 (53.3-64.7) | 18.5 (12.7-25.7) | 99.4 (96.9-100) | 2.4 (2.0-2.8) | 0.1 (0.01-0.4) |
|   | ≥ 4 | High | 28 (96.5) | 123 (40.9) | | | | | | |
| 5 | < 5 | Low | 5 (17.2) | 220 (73.1) | 82.8 (64.2-94.2) | 73.1 (67.7-78.0) | 22.9 (15.2-32.1) | 97.8 (94.9-99.3) | 3.1 (2.4-3.9) | 0.2 (0.1-0.5) |
|   | ≥ 5 | High | 24 (82.7) | 81 (26.9) | | | | | | |
| 6 | < 6 | Low | 15 (51.7) | 277 (92.0) | 48.3 (29.4-67.5) | 92.0 (88.4-94.8) | 36.8 (21.8-54.0) | 94.9 (91.7-97.1) | 6.1 (3.5-10.3) | 0.6 (0.4-0.8) |
|   | ≥ 6 | High | 14 (48.3) | 24 (7.9) | | | | | | |

Moreover, our study found the significant association between RDW and PICU mortality. Similarly, the clinical utility of RDW has also been affirmed in numerous studies involving critically ill patients [9–13,25–27]. RDW emerges as a straightforward and routinely reported metric, incurring no additional effort or costs, thereby presenting an advantageous prognostic factor. Elevated RDW levels may result from any pathological process disrupting RBC production, leading to the release of more immature RBCs into circulation. Furthermore, a high RDW has consistently been linked to increased inflammatory markers such as CRP levels, erythrocyte sedimentation rate (ESR), and interleukin-6 (IL-6) [39]. IL-6 and IL-1β have been shown to directly impair RBC survival in circulation, induce RBC membrane deformity, and suppress erythrocyte maturation [10,40,41]. Established research confirms that inflammatory cytokines disrupt RBC maturation within the bone marrow through various mechanisms, including inhibiting the production of or response to erythropoietin, causing impairments in iron metabolism and shortening RBC survival. Consequently, these inflammatory mediators can result in the release of newer and larger reticulocytes into peripheral circulation, thereby increasing RDW levels [39–44].

In addition, high PDW levels were observed within our cohort. Elevated PDW values, indicative of accelerated platelet turnover and reflecting a wide range of platelet sizes, demonstrated significant variation between sepsis survivors and non-survivors upon admission. Consistent with previous studies [15,45,46], these findings showed the association between high PDW and adverse outcomes. The increased PDW suggests platelet heterogeneity originating from bone marrow megakaryocytes, possibly due to platelet immaturity and swelling in circulation [45,47]. Moreover, a recent report has shown that IL-6, an acute-phase response inflammatory cytokine, regulates megakaryocytic maturation, platelet production, and platelet size [48]. These mechanisms may contribute to the observed elevation in PDW among critically ill patients.

Our study has several significant limitations. First, the retrospective design of the study limited the range of data that could be collected. Additionally, because the research was conducted at a single center, the findings may not be widely applicable. Second, with only 29 non-survivors in our cohort, we were constrained to use a maximum of 4 variables for developing the prediction score. Third, since neither thrombocytopenia nor mechanical ventilation parameters were associated with PICU mortality in our cohort, the initial sample size calculation may not have been adequate. To address this result, we recalculated our sample size of 330 to verify the power of statistical significance (0.8) using two-sample means comparisons of RDW between non-survivors (19.2 ± 3.9%) and survivors (16.6 ± 3.4%), with an overall PICU mortality rate of 8.8%. Our recalculation showed a power of 0.87, which is sufficient to support the reliability of our findings. While our sample size calculation ensures precise estimation of the average risk, it does not guarantee the prevention of over-fitting or a small absolute difference between the adjusted and apparent R-squared values. Fourth, our study included patients with leukemia or solid tumors, as these patients often require intensive care due to septic shock or respiratory failure, reflecting their heightened vulnerability. We aimed to provide a comprehensive representation of the overall patient population and mortality patterns in our PICU. However, we acknowledge that the inclusion of these patients may introduce a potential bias, particularly concerning post-chemotherapy immunosuppression, which could impact clinical

**Table 5. Multivariable full model of factors that are associated with overall ICU mortality in critically ill children (n = 330).**

| Variables within 24-hour of admission | | Full model (n= 330) | |
| --- | --- | --- | --- |
| | | OR (95% CI) | *P-value* |
| Age < 24 months | | 1.01 (0.21-4.78) | 0.985 |
| Male | | 1.77 (0.79-3.96) | 0.165 |
| BMI (kg/m²) | | 1.04 (0.92-1.18) | 0.476 |
| Comorbidities | None | Reference | |
| | 1 | 1.79 (0.19-16.33) | 0.603 |
| | ≥ 2 | 5.31 (0.57-49.02) | 0.141 |
| RDW≥15.9% | | 3.67 (1.42-9.49) | 0.007 |
| WBC | | 1.00 (0.99-1.00) | 0.270 |
| Platelet count | | 0.99 (0.99-1.00) | 0.174 |
| MPV≥10.25 fl | | 1.75 (0.65-4.76) | 0.267 |
| PDW | < 10% | Reference | |
| | 10-14.9% | 1.59 (0.52-4.85) | 0.410 |
| | ≥15.0% | 4.64 (0.88-24.47) | 0.070 |
| Use of mechanical ventilation | | 1.59 (0.47-5.37) | 0.451 |
| Use of vasoactive drugs | | 5.78 (1.93-17.23) | 0.002 |
| Multi-organ failure | | 1.36 (0.50-3.70) | 0.545 |

outcomes. Fifth, our study utilized 28-day PICU mortality as the primary outcome, which may differ from overall PICU mortality in certain aspects. To address this, we performed a multivariable analysis using a full model to identify factors associated with overall ICU mortality, as presented in Table 5, for further consideration. Lastly, we did not monitor the follow-up of clinical parameters, which would have enabled a more thorough assessment of the relationship between our model and patient outcomes. Despite these limitations, it remains crucial to identify at-risk patients in order to deliver high-quality care within resource constraints. Utilizing clinical parameters and inexpensive, widely available blood tests can aid in optimal management and resource allocation prioritization. Therefore, we advocate for the validation of our model in external datasets with a larger proportion of patients. Moreover, further prospective studies are warranted to validate our newly developed score and refine its clinical predictive accuracy.

## Conclusions

The ESCIC score, which integrates variables such as sex, vasoactive drug use, RDW, and PDW, demonstrates reliable and satisfactory predictive performance for mortality in critically ill children. This tool effectively identifies high-risk patients who may benefit from targeted interventions, thereby facilitating the delivery of high-quality care. However, external validation is necessary to confirm its generalizability.

## Supporting information

**S1 File. Minimaldataset_ESCIC.**
(XLS)

## Author contributions

**Conceptualization:** Kanokkarn Sunkonkit, Chatree Chai-adisaksopha, Rungrote Natesirinilkul, Phichayut Phinyo, Konlawij Trongtrakul.

**Data curation:** Kanokkarn Sunkonkit.

**Formal analysis:** Kanokkarn Sunkonkit, Phichayut Phinyo, Konlawij Trongtrakul.

**Funding acquisition:** Kanokkarn Sunkonkit.

**Investigation:** Kanokkarn Sunkonkit.

**Methodology:** Kanokkarn Sunkonkit, Phichayut Phinyo, Konlawij Trongtrakul.

**Project administration:** Kanokkarn Sunkonkit.

**Resources:** Kanokkarn Sunkonkit.

**Software:** Kanokkarn Sunkonkit, Phichayut Phinyo, Konlawij Trongtrakul.

**Supervision:** Chatree Chai-adisaksopha, Rungrote Natesirinilkul, Phichayut Phinyo, Konlawij Trongtrakul.

**Validation:** Kanokkarn Sunkonkit, Chatree Chai-adisaksopha, Rungrote Natesirinilkul, Phichayut Phinyo, Konlawij Trongtrakul.

**Visualization:** Kanokkarn Sunkonkit, Chatree Chai-adisaksopha, Rungrote Natesirinilkul, Phichayut Phinyo, Konlawij Trongtrakul.

**Writing – original draft:** Kanokkarn Sunkonkit, Konlawij Trongtrakul.

**Writing – review & editing:** Kanokkarn Sunkonkit, Chatree Chai-adisaksopha, Rungrote Natesirinilkul, Phichayut Phinyo, Konlawij Trongtrakul.

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
