## [Decision Letter · Decision Letter 0]

27 Jan 2025

PONE-D-24-47760Bedside Clinical Prediction Tool for Mortality in Critically Ill ChildrenPLOS ONE

Dear Dr. Trongtrakul,

Thank you for submitting your manuscript to PLOS ONE. After careful consideration, we feel that it has merit but does not fully meet PLOS ONE’s publication criteria as it currently stands. Therefore, we invite you to submit a revised version of the manuscript that addresses the points raised during the review process.

**ACADEMIC EDITOR: ** Daear Authors Please followe the reviewers' comments and  resubmitt a revised paper.Kind regards

We look forward to receiving your revised manuscript.

Kind regards,

Stefan Grosek, Ph.D., M.D.,

Academic Editor

PLOS ONE

Journal Requirements:

 “This work was supported by the Faculty of Medicine Research Fund, Chiang Mai University, Chiang Mai, Thailand (Grant No. 119/2566).”

Additional Editor Comments:

Dear Authors

Two reviewers finished their reviews and recommended minor changes in you manuscript.

Please follow their recommendtions and submitt a revised paper.

Kind regards

Reviewers' comments:

Reviewer's Responses to Questions

**Comments to the Author**

1. Is the manuscript technically sound, and do the data support the conclusions?

Reviewer #1: Yes

Reviewer #2: Partly

2. Has the statistical analysis been performed appropriately and rigorously? 

Reviewer #1: Yes

Reviewer #2: Yes

3. Have the authors made all data underlying the findings in their manuscript fully available?

Reviewer #1: Yes

Reviewer #2: Yes

4. Is the manuscript presented in an intelligible fashion and written in standard English?

Reviewer #1: Yes

Reviewer #2: Yes

5. Review Comments to the Author

Reviewer #1: The authors describe a new mortality score, developed from clinical data and hemogram, applicable in critically ill children. This score was internally validated using simulations, but was not externally validated.

The methods are robust, and the statistical analysis are well described.

The results are also well written, with figures and tables as appropriate. The conclusion fits the results and data.

However, i have few minor comments :

- the authors used the PIM-2 score. Why did not they chose the PIM-3 score, that is most commonly used ?

- It could be interesting to show the number of organ dysfunctions

- in other papers, the propensity score can be a good method to develop severity or mortality score. Did the authors envision to use this method for this manuscript ? if yes or no, why did not they use it ?

- the authors use the 28 days mortality as the outcome. did the authors try the overall mortality ? did it change the results ? i think it could be interesting to show the results with the overall mortality.

Reviewer #2: I thank the authors for their work. Accurate mortality risk prediction is indeed a priority in PICUs, in order to achieve adequate care level for each patient. The objective of enabling quick and efficient decision-making in primary-care hospitals is particularly interesting.

Line 113 : about exclusion criteria : what about children with malignant hematological diseases, in particular with post-chemotherapy aplasia ? I understand that you have patients with leukemia and soild tumor diagnoses in your cohort ? This probably introduces a bias in your results, with or without post-chemotherapy status. It may preferable to consider a malignant condition as an exclusion criterion.

Table 1 : you should integrate vasoactive inotropic score values, it would be more precise

Lines 233-234 : « with boys generally perceived as being biologically weaker and more susceptible to diseases ». The term « biological weakness » does not refer to scientific reliable data, it should not be used.

The references 40-41-42 are out of date, they should not be cited

Lines 236-237 : reference 43 : result is not significant. Reference 44 : diagnoses at admission profile differ from your unit. Study of reference 43 was conducted in a European Western country (Spain), diagnostic profiles are not comparable, and the sample size is quite small.

To be honest, lines from 231 to 242 are of poor interest.

Lines 296-300 (conclusions) : you should insist on the need for external validation

6. PLOS authors have the option to publish the peer review history of their article (what does this mean?). If published, this will include your full peer review and any attached files.

Reviewer #1: No

Reviewer #2: **Yes: **Charlie DE MELO

---

## [Author Response · Author response to Decision Letter 1]

13 Feb 2025

Dear Editor and Reviewers,

Thank you very much for your comments. We appreciate the time and effort by the editors and reviewers in reviewing this manuscript. Your comments and suggestions have been taken into consideration. We have incorporated the necessary suggestions in the attached version of the manuscript. Please find below, a point-by-point reply to each of your comments. Thank you in advance for your consideration.

Reviewer #1:

The authors describe a new mortality score, developed from clinical data and hemogram, applicable in critically ill children. This score was internally validated using simulations, but was not externally validated.

The methods are robust, and the statistical analysis are well described.

The results are also well written, with figures and tables as appropriate. The conclusion fits the results and data.

However, I have few minor comments:

1. The authors used the PIM-2 score. Why did not they choose the PIM-3 score, that is most commonly used?

Response: Thank you for your insightful comment. This study is retrospective, collecting data from January 2018 to December 2022. During this period, the PIM-2 score was the standard mortality risk assessment tool used in our PICU and was integrated into our hospital’s automated data recording system. While the PIM-3 score has since become more widely utilized, the transition from PIM-2 to PIM-3 within our system is still in progress. As such, the data available for comparison with the new ESCIC score in this study was based on the PIM-2 score, reflecting the routine practice in our setting during the study period. We appreciate your understanding of this context.

2. It could be interesting to show the number of organ dysfunctions.

Response: Thank you for your valuable comment. We have included the number of multiorgan failures and found that the non-survivor group had a significantly greater number of organ failures compared to the survivor group (4 [0, 4] vs. 0 [0, 1]; OR 4.04 [2.69–6.04], p<0.001). Accordingly, we have incorporated these data into Table 1.

3. In other papers, the propensity score can be a good method to develop severity or mortality score. Did the authors envision to use this method for this manuscript? if yes or no, why did not they use it?

Response: Thank you for your insightful question. While propensity score methods are indeed valuable for developing severity or mortality prediction scores, we did not employ this approach in our study due to the relatively limited sample size (n = 330). Propensity score methods typically require a larger cohort to ensure robust estimation and effective covariate balancing, particularly when incorporating multiple predictors into logistic regression models. Given these considerations, we opted for multivariable logistic regression, which is more suitable for smaller datasets and allows for a direct assessment of the associations between predictors and mortality. We appreciate your valuable suggestion, and the potential application of propensity score methodology will be considered in future research endeavors. Thank you.

4. The authors use the 28 days mortality as the outcome. Did the authors try the overall mortality? Did it change the results? I think it could be interesting to show the results with the overall mortality.

Response: Thank you for your insightful comment. We have analyzed the data using overall mortality as the outcome measure. Among the 330 children, 37 (11.21%) died in the PICU. Additionally, we conducted a multivariable analysis using a full model to identify factors associated with overall ICU mortality, as presented in Table 5. We have also incorporated this information into the discussion section (limitations), revised the title of Table 2, and added Table 5 accordingly.

Discussion (page 16, line 286-290)

“Fifth, our study utilized 28-day PICU mortality as the primary outcome, which may differ from overall PICU mortality in certain aspects. To address this, we performed a multivariable analysis using a full model to identify factors associated with overall ICU mortality, as presented in Table 5, for further consideration.”

Table 2 (page 29, line 517-518)

Multivariable full model of factors that associated with 28-day ICU mortality in critically ill children (n = 330)

Table 5 (page 32, line 524-525)

Multivariable full model of factors that associated with overall ICU mortality in critically ill children (n = 330)

Reviewer #2:

I thank the authors for their work. Accurate mortality risk prediction is indeed a priority in PICUs, in order to achieve adequate care level for each patient. The objective of enabling quick and efficient decision-making in primary-care hospitals is particularly interesting.

1. Line 113: about exclusion criteria: what about children with malignant hematological diseases, in particular with post-chemotherapy aplasia? I understand that you have patients with leukemia and soild tumor diagnoses in your cohort? This probably introduces a bias in your results, with or without post-chemotherapy status. It may preferable to consider a malignant condition as an exclusion criterion.

Response: Thank you for your insightful comment. Our cohort includes patients with leukemia or solid tumors, comprising 17 out of 330 patients (5%) admitted to the PICU. The majority of these patients required intensive care due to septic shock or respiratory failure, and 4 of the 17 (23.5%) were in the non-survivor group, highlighting their vulnerability. Excluding these patients could limit the generalizability of our findings and may not accurately reflect the overall patient population and mortality patterns in our PICU. We acknowledge that the inclusion of these patients may introduce a potential bias. To address this, we have explicitly noted this as a study limitation in the discussion section.

Discussion (page 16, line 281-286)

“Fourth, our study included patients with leukemia or solid tumors, as these patients often require intensive care due to septic shock or respiratory failure, reflecting their heightened vulnerability. We aimed to provide a comprehensive representation of the overall patient population and mortality patterns in our PICU. However, we acknowledge that the inclusion of these patients may introduce a potential bias, particularly concerning post-chemotherapy immunosuppression, which could impact clinical outcomes.”

2. Table 1: you should integrate vasoactive inotropic score values, it would be more precise.

Response: Thank you for your insightful comment. We have calculated the vasoactive inotropic score within the first 24 hours post-admission and found that the non-survivor group had a significantly higher score compared to the survivor group (120 [10, 150] vs. 0 [0, 10]; OR 1.06 [1.03–1.07], p<0.001). Accordingly, we have incorporated the vasoactive inotropic score values into Table 1.

3. Lines 233-234: « with boys generally perceived as being biologically weaker and more susceptible to diseases ». The term « biological weakness » does not refer to scientific reliable data, it should not be used.

Response: Thank you for your comment. We have edited the paragraph in the discussion.

Discussion (page 13, line 227-236)

“In our study, we observed a higher mortality rate among male patients in the PICU. This finding aligns with existing literature that suggests gender differences in critical care outcomes [37-39]. However, the reasons behind increased mortality in male PICU patients are multifaceted. Biological factors, such as differences in immune response between genders, may play a significant role. Additionally, behavioral and social factors could contribute to the observed disparities. It's important to note that findings on gender differences in PICU mortality are not entirely consistent across studies. Given these complexities, further research is necessary to elucidate the underlying mechanisms contributing to gender disparities in PICU mortality. Understanding these factors is crucial for developing targeted interventions aimed at improving outcomes for all critically ill children.”

4. The references 40-41-42 are out of date, they should not be cited.

Response: Thank you for your comment. We have removed the previous references (40–42) and revised the discussion accordingly (page 13, lines 227–236).

5. Lines 236-237: reference 43: result is not significant. Reference 44: diagnoses at admission profile differ from your unit. Study of reference 43 was conducted in a European Western country (Spain), diagnostic profiles are not comparable, and the sample size is quite small.

To be honest, lines from 231 to 242 are of poor interest.

Response: Thank you for your suggestion. We have removed the previous references (43–44) and revised the corresponding paragraph in the discussion section.

Discussion (page 13, line 227-236)

“In our study, we observed a higher mortality rate among male patients in the PICU. This finding aligns with existing literature that suggests gender differences in critical care outcomes [37-39]. However, the reasons behind increased mortality in male PICU patients are multifaceted. Biological factors, such as differences in immune response between genders, may play a significant role. Additionally, behavioral and social factors could contribute to the observed disparities. It's important to note that findings on gender differences in PICU mortality are not entirely consistent across studies. Given these complexities, further research is necessary to elucidate the underlying mechanisms contributing to gender disparities in PICU mortality. Understanding these factors is crucial for developing targeted interventions aimed at improving outcomes for all critically ill children.”

6. Lines 296-300 (conclusions): you should insist on the need for external validation

Response: Thank you for your valuable suggestion. We have emphasized the necessity of external validation in the conclusion section.

Conclusions (page 16, line 299-303)

“The ESCIC score, which integrates variables such as sex, vasoactive drug use, RDW, and PDW, demonstrates reliable and satisfactory predictive performance for mortality in critically ill children. This tool effectively identifies high-risk patients who may benefit from targeted interventions, thereby facilitating the delivery of high-quality care. However, external validation is necessary to confirm its generalizability.”

Best Regards,

Konlawij Trongtrakul, MD, PhD

---

## [Decision Letter · Decision Letter 1]

17 Mar 2025

Bedside Clinical Prediction Tool for Mortality in Critically Ill Children

PONE-D-24-47760R1

Dear Dr. Trongtrakul,

We’re pleased to inform you that your manuscript has been judged scientifically suitable for publication and will be formally accepted for publication once it meets all outstanding technical requirements.

Kind regards,

Stefan Grosek, Ph.D., M.D.,

Academic Editor

PLOS ONE

Additional Editor Comments (optional):

Dear Authors

Your vrticle "Bedside prediction tool for mortality in critically ill children" is very interesting study which brings to the readers new interesting prediction tool for mortality derived from four known indicators but not yet included together in prediction model (male gender, use of vasoactive drugs, red blood, cell distribution width (RDW) ≥15.9%, and platelet distribution width (PDW), categorized as follows: <10% (0 points), 10-14.9% (2 points), and ≥15% (4 points)).

All comments raised by the reviewers were addressed and they also recommend as my self to accept this article for publication.

Reviewers' comments:

Reviewer's Responses to Questions

**Comments to the Author**

1. If the authors have adequately addressed your comments raised in a previous round of review and you feel that this manuscript is now acceptable for publication, you may indicate that here to bypass the “Comments to the Author” section, enter your conflict of interest statement in the “Confidential to Editor” section, and submit your "Accept" recommendation.

Reviewer #1: All comments have been addressed

Reviewer #2: All comments have been addressed

2. Is the manuscript technically sound, and do the data support the conclusions?

Reviewer #1: Yes

Reviewer #2: Yes

3. Has the statistical analysis been performed appropriately and rigorously? 

Reviewer #1: Yes

Reviewer #2: Yes

4. Have the authors made all data underlying the findings in their manuscript fully available?

Reviewer #1: Yes

Reviewer #2: Yes

5. Is the manuscript presented in an intelligible fashion and written in standard English?

Reviewer #1: Yes

Reviewer #2: Yes

6. Review Comments to the Author

Reviewer #1: The authors have addressed all the comments made by the reviewer.

The manuscript is ready to be accepted from the reviewer's point of view.

Reviewer #2: I thank the authors for considering my reviewing.

I find this work interesting, with potentially valuable impact on mortality prediction accuracy in PICUs.

Kind regards,

Charlie de Melo, M.D.

7. PLOS authors have the option to publish the peer review history of their article (what does this mean?). If published, this will include your full peer review and any attached files.

Reviewer #1: No

Reviewer #2: **Yes: **Charlie de Melo

---

## [Editor Report · Acceptance letter]

PONE-D-24-47760R1

PLOS ONE

Dear Dr. Trongtrakul,

I'm pleased to inform you that your manuscript has been deemed suitable for publication in PLOS ONE. Congratulations! Your manuscript is now being handed over to our production team.

Kind regards,

on behalf of

Professor Stefan Grosek

Academic Editor

PLOS ONE
